# Circulating Tumor Cells Enumeration from the Portal Vein for Risk Stratification in Early Pancreatic Cancer Patients

**DOI:** 10.3390/cancers13246153

**Published:** 2021-12-07

**Authors:** Javier Padillo-Ruiz, Gonzalo Suarez, Sheila Pereira, Francisco José Calero-Castro, Jose Tinoco, Luis Marin, Carmen Bernal, Carmen Cepeda-Franco, Jose Maria Alamo, Francisco Almoguera, Hada C. Macher, Paula Villanueva, Francisco José García-Fernandez, Inmaculada Gallego, Manuel Romero, Miguel Angel Gomez-Bravo, Valeria Denninghoff, María José Serrano

**Affiliations:** 1Instituto de Biomedicina de Sevilla, University Hospital Virgen del Rocío, 41013 Seville, Spain; gsartacho@gmail.com (G.S.); spereira-ibis@us.es (S.P.); fjcalerocastro@gmail.com (F.J.C.-C.); jose.tinoco.gonzalez@gmail.com (J.T.); marinlm@hotmail.com (L.M.); cbernalb@hotmail.com (C.B.); carmencepedafranco@gmail.com (C.C.-F.); jmalamom@hotmail.com (J.M.A.); Fcoal94@gmail.com (F.A.); paulavg1995@gmail.com (P.V.); miagb@msn.com (M.A.G.-B.); 2Department of Molecular Biochemistry, University Hospital Virgen del Rocío, 41013 Seville, Spain; hadacmacher@icloud.com; 3Department of Gastroenterology, University Hospital Virgen del Rocío, 41013 Seville, Spain; fjgarciaf.hvrocio@gmail.com (F.J.G.-F.); mromerogomez@us.es (M.R.); 4Department of Oncology, University Hospital Virgen del Rocío, 41013 Seville, Spain; inmagallego84@hotmail.com; 5Molecular-Clinical Lab, University of Buenos Aires (UBA)—National Council for Scientific and Technical Research (CONICET), Buenos Aires C1122AAH, Argentina; 6Oncology Unit, Centre for Genomics and Oncological Research-GENYO, Pfizer, University of Granada, Andalusian Regional Government, 18016 Granada, Spain; mjose.serrano@genyo.es; 7Integral Oncology Division, Instituto Biosantario Granada (iBS-Granada), Virgen de las Nieves University Hospital, 18012 Granada, Spain; 8Department of Pathological Anatomy, Faculty of Medicine, University of Granada, 18071 Granada, Spain

**Keywords:** circulating tumor cell, cluster, portal vein, central venous catheter, pancreatic cancer, early stage, death risk stratification

## Abstract

**Simple Summary:**

Effective biomarkers are needed to enable personalized medicine for pancreatic cancer patients. This study analyzes the prognostic value, in early pancreatic cancer, of circulating tumor cells and clusters from the central venous catheter and portal blood. Circulating tumor cells were isolated using an immunomagnetic selection and were detected by microscopy using immunocytochemistry staining. In conclusion, the circulating tumor cell number in portal blood identifies a death risk in patients with early pancreatic cancer.

**Abstract:**

Background. Effective biomarkers are needed to enable personalized medicine for pancreatic cancer patients. This study analyzes the prognostic value, in early pancreatic cancer, of single circulating tumor cell (CTC) and CTC clusters from the central venous catheter (CVC) and portal blood (PV). Methods. In total, 7 mL of PV and CVC blood from 35 patients with early pancreatic cancer were analyzed. CTC were isolated using a positive immunomagnetic selection. The detection and identification of CTC were performed by immunocytochemistry (ICC) and were analyzed by Epi-fluorescence and confocal microscopy. Results. CTC and the clusters were detected both in PV and CVC. In both samples, the CTC number per cluster was higher in patients with grade three or poorly differentiated tumors (G3) than in patients with well (G1) or moderately (G2) differentiated. Patients with fewer than 185 CTC in PV exhibited a longer OS than patients with more than 185 CTC (24.5 vs. 10.0 months; *p* = 0.018). Similarly, patients with fewer than 15 clusters in PV showed a longer OS than patients with more than 15 clusters (19 vs. 10 months; *p* = 0.004). These significant correlations were not observed in CVC analyses. Conclusions. CTC presence in PV could be an important prognostic factor to predict poor prognosis in early pancreatic cancer. In addition, the number of clustered-CTC correlate to a tumor negative differentiation degree and, therefore, could be used as a diagnostic biomarker for pancreatic cancer.


**Highlights**
CTCs can be detected in the early stages of pancreatic cancer.CTC inside a cluster is much higher in G3 than in G1–2 in both samples.The larger the size of the tumor, the greater the number of total CTC for PV and CVC samples.The number of CTC < 185 in PV (HR = 4.464; *p* = 0.016) and no vascular invasion (HR: 3.663; *p* = 0.013) were independent predictors of better long-term survival.


## 1. Introduction

Pancreatic ductal adenocarcinoma (PDAC) is one of the most aggressive cancers associated with poor prognosis and high mortality. It has a natural tendency toward very early spread, even in resectable cases. Low-grade pancreatic cancers (G1) tend to grow and spread more slowly than high-grade (G3) cancers. Most of the time, G3 tends to have a poor prognosis compared to G1 or G2 cancers [1,2,3]. Therefore, it is necessary to find prognostic markers that identify the minimal residual disease and predict the risk of relapse. Seriate analyses of CA19-9 during chemotherapy represent practical and specific markers of response to the treatment itself. Other relevant prognostic parameters are the presence of perineural, vascular, and lymphatic invasion, which are essential characteristics for tumor growth and dissemination [1,2,3].

Unfortunately, there are currently no prognostic and predictive markers to indicate the presence of vascular, neural, or lymphatic invasion, hindering the clinical management of patients and, therefore, their possibility of cure. Among several biomarkers that could be evaluated preoperatively, single circulating tumor cells (CTC), as part of the liquid biopsy family, could be one of the proposed clinical markers. Several studies have reported CTC’s prognostic and predictive value in cancer patients, including pancreatic cancer [4,5,6]. In solid tumors, detection of CTC is usually performed using peripheral blood. Still, this search has not been possible in pancreatic cancer because blood flow first drains into the portal vein (PV), continues to the liver, and finally reaches the peripheral blood. Consistently, previous work has shown that CTC in patients with pancreatic cancer is detected with greater certainty in PV than in the peripheral venous blood of such patients [7]. Little is known about the characteristics and clinical implications of finding CTC clusters in PV. Although there are studies in advanced pancreatic cancer [8,9], only a few evaluate the role of CTC and clusters in the portal and central venous blood in patients with early pancreatic cancer as preoperative markers of risk stratification [7,10]. In this work, CTC and clusters were evaluated in patients with early pancreatic adenocarcinoma from the portal and central venous blood samples collected simultaneously during pancreaticoduodenectomy (PD). This study aimed to correlate CTC and clusters count from PV and central venous catheter (CVC) with tumor grade, preoperatively tumor size/CA19-9, and vascular/lymphatic/neural invasion. Additionally, to correlate this finding with overall survival (OS) and disease-free interval (DFS), including liver metastasis or local recurrence.

## 2. Material and Methods

### 2.1. Study Design and Sample Collection

This work was a prospective cross-sectional and longitudinal study involving 35 patients with carcinoma in the head of the pancreas. Patients with metastasis were excluded, and all patients received gemcitabine as treatment. Twenty-two were men and 13 women with a mean age of 67.4 ± 9.8 years old. The Ethics Committee of the University Hospital Virgen del Rocio approved the study protocol. Informed consent was obtained from all patients. According to the American Joint Committee on Cancer (AJCC) TNM Classification of Malignant Tumors, the study included stages I (*n* = 15) and II (*n* = 20) of disease. Only patients in whom no metastasis was shown by 18F-Fluro-deoxyglucose positron emission tomography and computed tomography (FDG-PET-CT) underwent surgery. Biopsy was also performed for any suspicious lesions, and if an intraoperative analysis was positive for metastatic adenocarcinoma, PD was canceled. The exclusion criteria were metastatic cancer diagnosis, locally advanced tumor, pancreatitis episodes within the three months before surgery, and previous history of cancer within the past five years. The tumor size and CA 19-9 (268.3 ± 586.9 U/mL) were preoperatively evaluated. PD was performed in all 35 patients. Complete resection (R0) was achieved in 24 (69%) patients according to established protocol [11,12]. Based on their differentiation, tumors were divided into three grades: well-differentiated (G1) in 8 (23%) patients, moderately differentiated (G2) in 20 (57%), and poorly differentiated (G3) in 7 (20.0%). The microscopic vascular invasion was observed in 13 (37%) patients, lymphatic invasion in 9 (26%), and neural invasion in 18 (51%). Adjuvant chemotherapy based on gemcitabine-based formula was administered postoperatively after pathological diagnosis of pancreatic carcinoma for 35 patients. All cases were followed prospectively during 24 months. A CT scan was performed to diagnose liver metastases and local recurrence as a low-density mass in the liver or surgical area.

### 2.2. Validation of CTC Detection

Cell culture Pancreatic cancer cell PANC-1 (ATCC^®^ CRL-1469, Manassas, VA, USA) were cultured in Dulbecco’s Modified Eagle’s Medium (DMEM, Lonza BE12-604F, Basel, Switzerland) supplemented with 10% fetal bovine serum (F7524, Sigma-Aldrich, St. Louis, MO, USA) and 1% Penicillin/Streptomycin (15140122, GIBCO). Cell cultures were grown in a humidified incubator with 5% CO_2_ at 37 °C. Once the cells were needed, the cells were washed with Trypsin-EDTA (25300062, GIBCO, Thermo Fisher Scientific, Inc., Waltham, MA, USA). Pancreatic cancer cells (PCC) were used as a positive control. Two control groups were used for negative and positive control. For technical negative control, 7 mL aliquots of peripheral blood were collected from healthy donors (*n* = 3) and were mixed with approximately 300 PCC for positive control. All samples were isolated with the Isoflux™ system and detected following the manufacturer’s protocol by Fluxion. The specificity of the anti-cytokeratin (CK) was evaluated using the pancreatic tumor PANC-1 cell line (ATCC^®^ CRL-1469, Manassas, VA, USA) as a positive control because it has a high CK protein expression; and the peripheral blood mononuclear cell (PMBC) as a negative control for its non-expression of EpCAM and CK (source: The human protein atlas).

Clinical negative controls of PV samples would not be ethically approved because there are surgical risks in that sampling. Regardless of this group, we included CVC samples from a non-neoplastic control group based on their clinical history, such as those undergoing cardiac surgeries. Therefore, the sample from 8 patients was extracted from the pre-implanted CVC before cardiac surgery. We performed oncological biochemical markers of these patients in the sample extracted simultaneously: prostate-specific antigen (PSA), alpha-fetoprotein (AFP), carcinoembryonic antigen (CEA), carbohydrate antigen-15.3 (CA15-3), carbohydrate antigen-19.9 (CA-19.9), carbohydrate antigen-125 (CA-125). As an inclusion criterion to consider as proper negative controls for the recruited patients, we determined that the oncological markers tested in the sample extracted simultaneously and in the same place as the one extracted for CTC evaluation had average values.

### 2.3. CTC Isolation, Detection, and Enumeration

During surgery, before manipulating the tumor, 7 mL blood samples were simultaneously obtained from the CVC tip in the superior vena cava and from the PV by direct punction. The first blood-draw was discarded to exclude epithelial cells dislodged by the venipuncture (these factors play a role in the frequency of epithelial cells in the blood, as these steps may lead to unspecific shedding of epithelial cells). Both patient samples were collected in K2-EDTA Vacutainer tubes, maintained at room temperature, and processed 24 h after collection. Blood samples were enriched in peripheral mononuclear blood cells using gradient centrifugation with Histopaque^®^-1119, and CTC were isolated using the IsoFlux platform. Isoflux platform has been designed to isolate the CTC based on a microfluidic process for immunomagnetic positive selection. The IsoFlux™ system utilizes micrometer-scale beads, which have been shown to result in a magnetic moment that is sufficient for capturing cells even with low target expression [13]. The process is automatic, increasing the ability to capture the CTC. The Isoflux™ Epithelial to Mesenchymal Transitions Circulating Tumor Cell Enrichment Kit (EMT Enrichment Kit, Izasa, Catalog N.910-0106, Werfen, San Diego, CA, USA) was used for performing CTC enrichment. In this kit, beads were conjugated with four different antibodies, targeting both epithelial and mesenchymal markers. The kit utilizes both anti-EpCAM and anti-EGFR-antibodies for the detection of epithelial cells, as well as anti-N-Cadherin and anti-Vimentin as mesenchymal markers. The EpCAM is a cell surface molecule known to be highly expressed in solid cancers. The immunomagnetic beads conjugated with antibodies were added to cells suspended in IsoFlux Binding Buffer and incubated for 90 min at 4 °C, as indicated in the Fluxion protocol. Subsequently, they were subjected to immunomagnetic isolation with IsoFlux (Fluxion Biosciences Inc). After the sample was processed, the enriched cells were fixed and stained with the fluorescent reagents (Isoflux™ Circulating Tumor Cell Enumeration Kit Izasa Catalog N.910-0093, Werfen, San Diego, CA, USA). The fluorescent reagents included were anti-CK-fluorescein isothiocyanate (FITC) specific for the intracellular protein CK, characteristic of epithelial cells; anti-CD45-Indocarbocyanine (Cy3) specific for leukocytes; and Hoechst 33342, a nucleic acid stain cell-permeant nuclear counterstain that emits blue fluorescence when bound to AT-rich regions of the minor groove in DNA. The CTC detection and enumeration were performed by fluorescence microscopy, and the images obtained were processed using image software based on the Hough transform (VR-CTC). We had described previously a method based on image processing to count sets of pixels showing a cell nucleus, cytokeratin expression, and no CD45 expression [14]. To count them, the Hough transform was used. The approach allowed a classification of the events counted in CTC, cluster, and clustered-CTC. Then, CTC were enumerated as morphologically intact CK+/CD45-/nucleated cells. The size of cells and clusters and the number of clustered CTC were also characterized. The mean coefficient of variation in CTC determination was less than 2% [0–1.41]. The sensitivity presented by this approach was 85.58%, and the specificity was 88.16% [14]. The results obtained were compared with counting undertaken by a technician. High correlations were demonstrated in total pancreatic tumor cells in healthy donors (R^2^ = 0.995) and patients with pancreatic cancer (R^2^ = 0.955) as well as in free cells (R^2^ = 0.993 and R^2^ = 0.975, respectively) [14].

### 2.4. Statistical Analysis

Single regression analysis was performed to assess the linearity of the two blood samples for detecting CTC. The correlation of CTC counts and tumor invasion (vascular, lymphatic, and neural) was analyzed using the Mann–Whitney U test and the Kruskal–Wallis H test. The distributions of patients above and below the cut-off level in CTC were compared using Fisher’s exact test. Spearman’s rank correlation coefficient examined the relationship between CTC counts and the tumor marker (Ca19.9). Survival analysis was done using the product-limit Kaplan–Meier and the log-rank test. Multivariate analysis was performed using a Cox proportional hazard model to test for independent prognostic variables. All statistical calculations were carried out using IBM Corp. released 2020, IBM SPSS Statistics for Windows, Version 27.0. Armonk, NY, IBM Corp. The required significance was *p* < 0.05.

## 3. Results

### 3.1. Validation of CTC Detection

These experimental analyses included tumor cell lines and healthy donors as positive and negative technical controls, respectively. Negative control presented 0.67 ± 0.31 CTC/mL in blood from healthy donors whereas positive control showed 37.15 ± 9.09 PCC/mL (PCC total = 185.75 ± 45.43). Positive controls showed that the detection rate of CTCs with this methodology is approximately 55%.

The non-neoplastic control group included eight CVC samples from patients who underwent cardiac surgeries, had no history of cancer in their medical history, and had average results in their tumor biochemical markers taken from the sample simultaneously. These negative controls showed 0.87 ± 0.39 CTC/mL in CVC blood, which found no clusters. The median obtained was 0.50 CTC/mL with a range between 0.00 and 1.75. There are differences in CVC between the number of CTC/mL of patients with pancreatic cancer and non-neoplastic patients who underwent cardiac surgery (U-Mann Whitney *p*-value < 0.001). No clusters were found in the non-neoplastic group, unlike in samples from pancreatic cancer patients with a median (range) of 14.5 (3.8–35.5) cluster/mL.

### 3.2. Longitudinal Enumeration and Cluster Evaluation

Single CTC and clusters were detected in all patients, both in PV and CVC (Figure 1). The analysis showing the correlation of CTC values on CVC vs. PV appeared in Table 1. In the three measures, Pearson’ s correlation had a coefficient r = 0.6 between PV and CVC. We found more CTC, clusters, and total CTC in CVC than PV, with no statistically significant differences (Table 1). The ratio and sizes are merely informative data used as quality control between trials, which should not differ between the samples 291.8 (120.0–500.0).

When the data obtained were compared by discriminating according to the tumor grade, the values reached in the same tumor grade showed that in CVC, they are higher than in PV (Table 2 and Table 3). When comparing tumor grades for the same sampling site, we found that for both samples, the data that most attracts attention is CTC inside a cluster, which is much higher in G3 than in G1/2, despite not having significant differences. The ratio and sizes have no variation as we expected.

The preoperative CA 19-9 levels and tumor size measured by CT scan were analyzed and correlated with CTC and CTC clusters. In both samples, CVC and PV, we did not find a significant association (Table 4 and Table 5). We used the Spearman correlation coefficient (Rho) for these analyses, which takes values from +1 to −1. A Rho of +1 indicates a perfect association of ranks, a Rho of zero indicates no association between levels, and an r of −1 indicates a perfect negative association of grades. The closer Rho is to zero, the weaker the association between the ranks. Therefore, relationships identified using correlation coefficients should be interpreted as associations and not as causal relationships [15]. In both samples, the value of CA 19-9 has a positive relationship with CTC/cluster.

Other factors related to survival are neural, vascular, and lymphatic invasion (Table 6 and Table 7). There was no significant correlation between CTC measurements with neural, vascular, or lymphatic invasion. Still, we are struck by the high number of CTCs (free and total) in PV samples in patients with vascular invasion concerning those who do not have it. The biological mechanism of tumor hematogenous dispersal is associated with vascular invasion.

### 3.3. Prognostic Factors for Long-Term Survival

Regarding OS, the median survival time was 18 months (range 12.5–23.5 months) without postoperative mortality. Thirty of the 35 patients (86%) survived for more than six months, with 20 (57%) of them still alive one year after the primary diagnosis.

Once the media (m) and standard derivation were calculated, based on Altmann optimal cut-off approximation, we estimated the p67 (or p33) value consulting in a probability table of the normal distribution the value of Pr (z < z) = 0.67, that corresponds to 0.724 [16]. Therefore, we estimate that quartile as m + 0.724s. Usually, the literature uses Q3 (p75) to consider the high value and Q1 (p25) to low values. We decided to use p33 instead of p25 to increase the sensibility at lower specificity. In our opinion, to evaluate possible cut-off points, we did not recommend using extreme values on each side. Therefore, it is excluded between 5 and 10% of each endpoint.

According to these results, to predict a G3 pancreatic adenocarcinoma presence, we established a cut-off for cluster number of 20 clusters/mL and CTC inside a cluster of 64 CTC/cluster. The prognostic value of CTC was set in the 33rd percentile (p33), resulting in a cut-off point of 185 (SD 45.3) CTC and 15 (SD 6.8) clusters [15].

Regarding PV measurements, the median patient’s survival was significantly longer in patients with less than 185 CTC (24.5 months (IC 19.6-29.4) vs. 10 months (CI 95% 7.4–12.5); *p* = 0.018) and less than 15 clusters (19 months (CI 95% 15.8–22.2) vs. 10 months (CI 95% 7.2–12.8); *p* = 0.004) (Table 8 and Figure 2). These findings were not found in CVC samples. Patients with vascular (*p* = 0.005) or lymphatic invasion (*p* = 0.044) were associated with less survival. However, the neural invasion did not correlate with OS (Figure 3).

To identify prognostic factors for long-term survival, we used a stratified univariate and multivariable Cox regression. The multivariable analysis showed that both the number of CTC < 185 (HR = 4.464; *p =* 0.016) and no vascular invasion (HR: 3.663; *p =* 0.013) were independent predictors of better long-term survival (Table 9).

### 3.4. Prognostic Factors for Long Term Local and Systemic Progression

Excluding incomplete resections, 23/35 (96%) patients had local recurrence disease-free survival for more than six months, with 22 (92%) of them being still local progression-free one year after the primary diagnosis. However, the median disease-free interval related to local recurrence was longer in patients with <185 CTC (28.5 vs. 25.0 months; *p* = 0.647) and less than 15 clusters (29.0 vs. 22.5 months; *p* = 0.787) in PV, differences were not significant. The vascular or neural invasion also failed to achieve correlation. Only the lymphatic invasion was associated with less local recurrence disease-free survival (12.0 vs. 23.0 months; *p* = 0.001). In fact, in the multivariable Cox regression analysis, the only lymphatic invasion was an independent predictor of disease-free interval for local recurrence (*p* = 0.031).

Regarding systemic progression (liver metastases), 18 of the 35 patients (75%) had systemic progression-free survival for more than one year after the primary diagnosis.

Regarding local recurrence, although the median disease-free interval related to metastases was longer in patients with <185 CTC (20.5 vs. 18.0 months; *p* = 0.636) and less than 15 clusters (24.0 vs. 18.0 months; *p* = 0.383) measured in PV, differences were not significant (Figure 4).

## 4. Discussion

In the present study, we analyzed the prognostic value of both single CTC and clustered-CTC presented in the samples of patients with early-stage adenocarcinoma in the head of the pancreas. One of the critical points in the therapeutic approach in pancreatic cancer is determining early markers that evaluate the prognosis and thus facilitate therapeutic decisions. At present, we must wait to have an appropriate biopsy to assess prognostic information. Biopsies are often difficult to obtain before deciding whether the best option for the patient is surgery or chemotherapy. Hence, the challenge is to be able to identify early markers in these patients. On the one hand, liquid biopsy and especially CTC have been postulated as potential biomarkers [17]. However, its determination in pancreatic cancer remains uncertain since different markers have been used to detect heterogeneous forms of CTC [10,18,19,20]. Additionally, on the other hand, clusters have been proposed as possible prognostic factors in pancreatic adenocarcinoma [17]. Nevertheless, the potential role of the number of cells within clusters or their size has not been determined. The present study determined the characteristics of both free CTC (number and size) and clusters (number, size, and CTC inside a cluster) by immunofluorescence with Hoechst 33342, epithelial marker cytokeratin, and CD45 through the Hough transform CTC measurement [14]. Samples were obtained from CVC and PV to determine the best approach to evaluate the potential correlation of CTC measurements with the patient’s prognosis.

CTC could be detected in the early stages of pancreatic cancer in both blood samples in all patients. CTC was listed as morphologically intact CK+/CD45-/nucleated cells [14]. As the positive immunomagnetic selection that has been used is not 100% effective in discriminating CD45 contaminant cells, if any, antibodies directed against CD45 were used to identify and differentiate CTCs from PBMC. Different combinations of devices and software for CTC enumeration have shown sensitivity, specificity, and a high correlation in the number of CTC detected, but always under the supervision of a technician. Our previous work described that the Hough transform was an excellent approach to counting CTC, defined as an EpCAM nucleated cell, positive CK, but CD45 negative. This image analysis method could be extrapolated to count CTC isolated by CellSearch^®^ or another device when samples are labeled EpCAM. In addition, this approach could allow us to count other cells by processing the image channels according to the expression of the spots that define the cells under study [14]. We chose the EpCAM-Isoflux™ System because, in previous works, significantly more CTC were isolated using the same isolation target, EpCAM, on the IsoFlux™ system, compared with CellSearch^®^. However, direct comparison of the CTC isolation by the two different platforms is complex due to using other clones of the same antibody and magnetic beads for capture. The IsoFlux™ system utilizes micrometer-scale beads, which have been shown to result in a magnetic moment that is sufficient for capturing cells even with low target expression [13]. Different detection sensibilities have been reported in the literature. Still, they are not comparable to our study because we have used a highly effective enrichment method with an automated form of enumeration, which has allowed us to detect CTCs in 100% of the samples in patients with early pancreatic cancer. The work of Hugenschmidt et al. studied CTC from blood samples before surgery, including patients with advanced stages with CellSearch^®^ enrichment, manual image analysis, and obtained 7% detection [21]. White et al. increased detection sensitivity to 71% because they studied CTC from PV samples and included patients with advanced stages with CellSearch^®^ enrichment and manual image analysis [22]. Buscail et al. detected 45–59% with two different CTC enrichment techniques using a ddPCR identification device for KRAS with a mutation rate of 92% in PDAC and including patients with advanced stages [23].

We found more CTC and clusters in the CVC samples than in the PV samples, opposed to all other previously published results and current understanding. However, the differences between the CTC detected in PV and CVC are not significant. As possible explanations, we propose first that the extraction of portal blood is done using fine needles; in contrast, CVC is done through the central route itself, leading to further destruction of cells in PV. Secondly, the CTC obtained in PV during surgery could be more susceptible to apoptosis; in contrast, central blood CTC, taking longer in the bloodstream, undergo phenotypic changes that would allow them to survive longer. Third, in PV, only newly discharged CTC were detected in the portal bloodstream, but in CVC, CTC could be due to previous discharges that remained in the bloodstream. However, as we have mentioned above, we found no statistically significant differences; therefore, we understand that this finding is not clinically relevant, unlike CTC levels’ impacts on PV associated with survival.

A case–control study would be impossible to perform because negative controls of PV samples would not be ethically approved because there are surgical risks in that sampling. To get an optimal surgical portal vein sample, a bile duct dissection, and transection, it is necessary; this dissection is only possible in selected surgeries such as duodenopancreatectomy, hepaticojejunostomy, or liver transplantation. Unfortunately, most of these procedures are done in patients with cancer or liver cirrhosis in whom hepatocarcinoma is presented. Still, we could not consider them true negative because CTC were likely related to hepatocarcinoma. On the other hand, regardless of this group, we included CVC samples from a non-neoplastic control group based on their clinical history, such as those undergoing cardiac surgeries. The control patients recruited had no oncology history, and the sample was extracted from the pre-implanted CVC before cardiac surgery. These patients had 0.87 ± 0.39 CTC/mL and neither cluster with the methodology performed.

CTC inside a cluster is much higher in G3 than in G1–2 in both samples. Once pancreatic cancer cells invade capillaries in the tumor tissue, they can enter portal veins for distal metastasis, such as metastasis to the liver and lung. Low-grade cancers (G1) tend to grow and spread more slowly than high-grade (G3) cancers. Most of the time, Grade 3 pancreas cancers tend to have a poor prognosis compared to Grade 1 or 2 cancers [1,2,3]. CTC clusters are highly metastatic. Acetato et al. have shown that CTC clusters in metastatic patients were related to shorter survival [24]. A cluster can include other cells, such as platelets, immune cells, and cancer-associated fibroblasts, providing a local microenvironment that protects CTC on the cluster and facilitates colonization [25]. Therefore, the quantification of the clustered-CTC could add relevant prognostic information.

In both samples, the value of CA 19-9 has a positive relationship with CTC/cluster. Spearman’s correlation coefficient allows us to observe that the number of total CTC is directly related to the size of the tumor. The more significant the cancer, the higher the total CTC number for both blood samples. However, a critical fact that allows us to glimpse the physiology of CTC is that clusters and the number of CTC of clusters have an indirect relationship for CVC and direct for PV. The blood sample in PV is taken closer to the single primary tumor; these findings would allow us to suggest that the clusters decrease in size in the journey from the portal vein, through the liver, to the bloodstream, decreasing the number of cells inside and releasing them as free CTC, therefore. However, the physical location of the CTC changes, the total number remains constant and in direct relation to the size of the tumor. As observed in other studies, in patients with early stages [26,27,28], none of the CTC evaluated parameters correlated with preoperative CA 19-9 or tumor size/TNM stage measured by CT scan either from CVC or PV. However, we found for the first time that the cluster number (>20 clusters/mL) and the CTC inside a cluster (>64 CTC) measured in PV correlated with a negative degree of tumor differentiation in the biopsy. These findings could guide the preoperative diagnosis when it is impossible to perform a preoperative biopsy.

Currently, the correlation of CTC values with progression-free survival or hepatic metastases in non-advanced pancreatic adenocarcinoma remains controversial [7,26,27,28,29,30]. For the survival and disease-free period analysis in our study, the cut-off was adjusted to the 33rd percentile (p33), and then we determined that the CTC´s cut-off was 185 cells, and cluster number’s cut-off [15]. As in other studies [7,31], the disease-free time in metastases appearance and local recurrences was lower in patients with values above the cut-off in CTC and clusters, although the differences were not statistically significant. Thus, we could not conclude that CTC and cluster levels are predictors of metastasis or local recurrence in the follow-up of patients in our study. In the multivariate analysis, CTC ≥ 185 in portal blood (HR 4.4) and vascular invasion (HR 3.6) were independent predictors for survival. However, related to OS, patients with more than 185 CTC and 15 clusters in PV had significantly lower survival. These findings were not observed in CVC determinations. White et al. have described that the number of CTC in PV correlates with survival. Still, this work has been carried out in different stages of the PDAC, with another isolation method and manual counting, which explains why they have found CTC in only 71% of the PV samples [22]. In coincidence with the work of White et al., the difference would be given by the methodology used since studies comparing CellSearch^®^ and Isoflux™ demonstrated that it is up to 8 times more sensitive, which would explain the high number [13]. Finding a 100% positivity rate, we thought these cells could be contaminated with epithelial cells, and therefore we performed thorough quality checks. However, the originality of our work is given in the clinical sensitivity that we have demonstrated through a cut-off that allows us to discriminate the forecast of the disease with some reliability. Consequently, the presence of false-positive endothelial cells would not be viable because CK is not expressed in these cells; therefore, we consider such a false positive to be unlikely.

There are no differences in PV between the number of CTC/mL and cluster/mL about the degrees of resection R0 and R1 (U-Mann Whitney *p*-value = 0.8556 and 0.637, respectably).

Patients with metastases were excluded, but patients who developed metastases during follow-up were analyzed. The study’s objective was to evaluate the CTC as part of an intraoperative liquid biopsy with a single non-disseminated tumor. By definition, a liquid biopsy must be obtained by non-invasive methods. Although getting blood from PV is clinically invasive, as all patients included in this study were susceptible to surgery as part of their treatment, we consider that the analyses of CTC from PV did not involve an additional invasive process.

Our study found that the determination of CTC and clusters in PV were better than in CVC as prognostic markers in patients with early-stage adenocarcinoma in the head of the pancreas. Therefore, cluster determinations and the number of CTC inside a cluster in PV could be helpful to assess the degree of differentiation of pancreatic carcinoma. The number of free CTC in PV would be beneficial to determine the long-term prognosis before the therapeutic decision. Given the low number of patients, this work is a hypothesis generator that determines that CTC presence in PV could be a prognostic factor to predict poor prognosis in early pancreatic cancer. To validate this proposal, these results need validation with a larger patient population and a longer follow-up.

## 5. Conclusions

CTC presence in PV could be a significant prognostic factor to predict poor prognosis in early pancreatic cancer. In addition, the number of CTC and clusters correlate to a tumor negative differentiation degree and, therefore, could be used as a diagnostic biomarker for early pancreatic cancer.

## Figures and Tables

**Figure 1 cancers-13-06153-f001:**
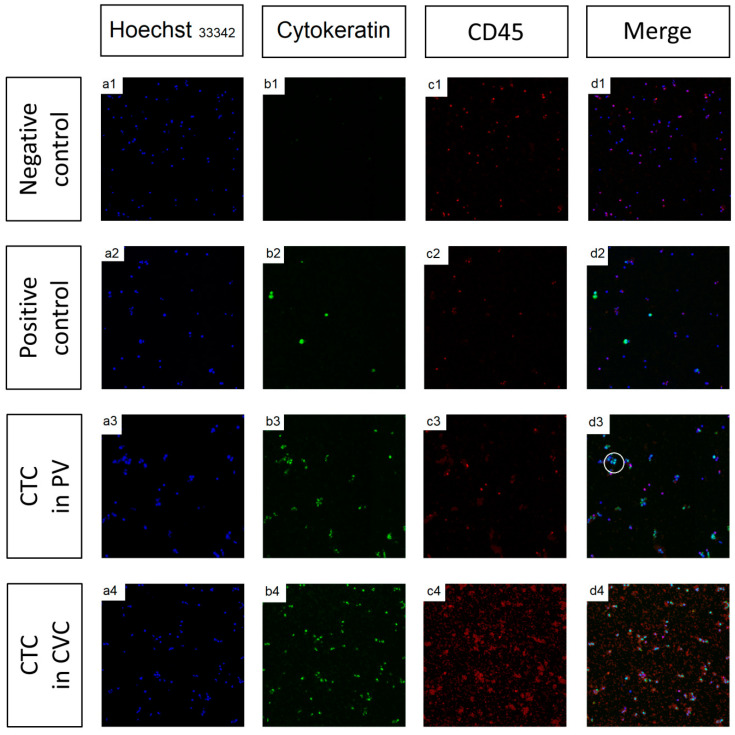
Epifluorescence microscopy pictures. The CTC detection and enumeration were performed by fluorescence microscopy, and the images obtained were processed using image software based on the Hough transform (VR-CTC). The CTC recovery was stained with: (**a**) anti-CK-fluorescein isothiocyanate (FITC) specific for the intracellular protein cytokeratin (characteristic of epithelial cells) (Green); (**b**) anti-CD45-Indocarbocyanine (Cy3) specific for leukocytes (Red); (**c**) Hoechst 33342, used for nuclear staining (Blue); and (**d**) merged image subject to logical operations where we performed a logical multiplication between the thresholding of channels Blue (nuclear) and Green (CK). Finally, we removed the pixels of the thresholding of channel Red (CD45). We eliminated artifacts from the resulting mask using Matlab tools. The white circle in d3 shows the merged image of CTC, which is CK+/CD45−/nucleated cells; in this way, image pixels are Hoechst+, CK+, and CD45—corresponding to a CTC. Data from (**1**) negative control (PMBC), (**2**) positive control (PANC-1 cell line), (**3**) CTC detected in PVC, and (**4**) CTC detected in CVC.

**Figure 2 cancers-13-06153-f002:**
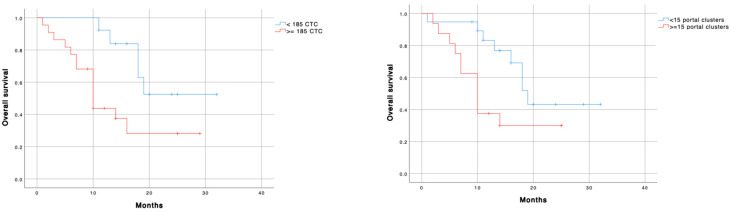
Overall survival according to left) PV-CTC’s cut-off; and right) PV-cluster number´s cut-off. Blue line: low cellular expression; Red line: high cellular expression. ≥185 CTC HR = 3.263 (1.135–5.221); *p* = 0.028. ≥15 clusters HR = 2.486 (0.989–6.247); *p* = 0.053.

**Figure 3 cancers-13-06153-f003:**
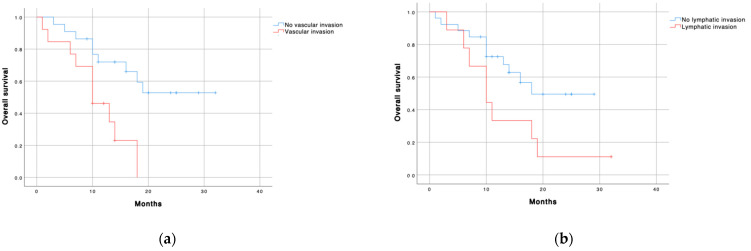
Overall survival according to (**a**) vascular invasion, HR = 3.568 (1.346–9.457); *p* = 0.011; (**b**) lymphatic invasion, HR = 2.418 (0.969–6.031); *p* = 0.058; (**c**) neural invasion, HR = 1.698 (0.666–4.328); *p* = 0.268; and (**d**) tumor grade differentiation, HR = 0.841 (0.196–3.771); *p* = 0.841. Blue line: no invasion or low grade (G1–2); red line: invasion or high grade (G3).

**Figure 4 cancers-13-06153-f004:**
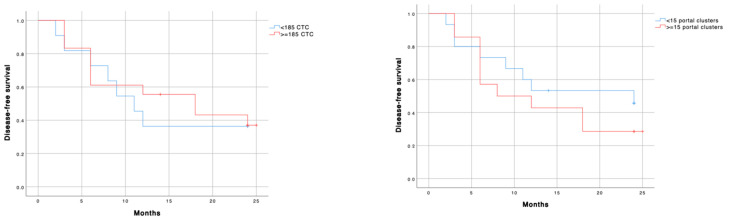
Disease-free survival according to left) PV-CTC´s cut-off; and right) PV-cluster number´s cut-off. Blue line: low cellular expression; red line: high cellular expression. <185 CTC (28.5 vs. 25.0 months; *p* = 0.647) and <15 clusters (29.0 vs. 22.5 months; *p* = 0.787).

**Table 1 cancers-13-06153-t001:** CTC characterization by Hough transform method.

Research Variable	PV (Median, Range)	CVC (Median, Range)	U-Mann Whitney *p*-Value	Correlation	Correlation *p*-Value
Free CTC (cell/mL)	235.4 (101.3–375.3)	291.8 (120.0–500.0)	0.151	0.6	0.004
Cluster/mL	12.9 (4.8–33.0)	14.5 (3.8–35.5)	0.622	0.6	0.001
CTC inside a cluster	30.4 (12.5–89.4)	37.4 (11.3–89.5)	0.205	0.5	0.008
Total CTC (cell/mL)	310.0 (132.1–446.0)	405.7 (130.7–553.8)	0.239	0.6	0.001
Ratio Free CTC/CTC inside a cluster	2.6 (2.4–3.0)	2.5 (2.3–2.9)	0.051	0.2	0.306
Free CTC median size	7.5 (6.8–8.7)	7.8 (6.7–9.0)	0.981	0.3	0.229
CTC inside a cluster median size	10.2 (8.5–11.6)	10.3 (8.6–11.7)	0.990	0.3	0.074

**Table 2 cancers-13-06153-t002:** Correlation CTC and clusters according to the degree of tumor differentiation (CVC measurements).

Research Variable	G1–G2	G3	*p*-Value
Free CTC (cell/mL)	279.2 (120.3–482.3)	357.0 (182.5–1020.8)	0.483
Cluster/mL	14.0 (3.3–31.7)	41.0 (17.9–46.8)	0.107
CTC inside a cluster	36.7 (8.7–73.7)	97.3 (54.5–116.3)	0.071
Total CTC (cell/mL)	399.0 (129.0–538.5)	481.0 (243.0–1117.7)	0.318
Ratio Free CTC/CTC inside a cluster	2.6 (2.4–2.9)	2.4 (2.2–3.1)	0.483
Free CTC median size	7.8 (6.6–9.1)	8.6 (6.7–10.2)	0.521
CTC inside a cluster median size	10.3 (8.7–11.6)	10.6 (7.8–13.2)	0.908

**Table 3 cancers-13-06153-t003:** Correlation CTC and clusters according to the degree of tumor differentiation (PV measurements).

Research Variable	G1–G2	G3	*p*-Value
Free CTC (cell/mL)	240.3 (102.2–373.3)	227.0 (140.8–559.0)	0.841
Cluster/mL	10.5 (4.3–32.7)	7.5 (30.0–80.3)	0.310
CTC inside a cluster	29.7 (9.2–74.2)	78.5 (21.3–260.6)	0.201
Total CTC (cell/mL)	314.5 (133.0–406.7)	305.5 (162.1–819.6)	0.725
Ratio Free CTC/CTC inside a cluster	2.6 (2.3–2.9)	2.8 (2.6–3.9)	0.150
Free CTC median size	7.3 (6.5–8.5)	8.5 (7.0–9.6)	0.310
CTC inside a cluster median size	9.7 (8.1–11.5)	10.2 (9.5–12.4)	0.335

**Table 4 cancers-13-06153-t004:** Tumor and CA 19-9 preoperatively evaluated by CTC characteristics with Spearman correlation coefficient (Rho) (CVC measurements).

Research Variable	CA 19-9 (Rho)	Tumor Size * (cm) (Rho)
Free CTC (cell/mL)	0.5	0.1
Cluster/mL	0.2	−0.1
CTC inside a cluster	0.2	−0.1
Total CTC (cell/mL)	0.4	0.4
Ratio Free CTC/CTC inside a cluster	−0.3	−0.3
Free CTC median size	−0.5	−0.1
CTC inside a cluster median size	−0.6	−0.6

* Measured by CT scan.

**Table 5 cancers-13-06153-t005:** Tumor and CA 19-9 preoperatively evaluated by CTC characteristics with Spearman correlation coefficient (Rho) (PV measurements).

Research Variable	CA 19-9 (Rho)	Tumor Size * (cm) (Rho)
Free CTC (cell/mL)	0.4	0.3
Cluster/mL	0.3	0.3
CTC inside a cluster	0.3	−0.1
Total CTC (cell/mL)	0.4	0.4
Ratio Free CTC/CTC inside a cluster	0	0
Free CTC median size	−0.1	−0.1
CTC inside a cluster median size	−0.2	0

* Measured by CT scan.

**Table 6 cancers-13-06153-t006:** Correlation of CTC and CTC clusters with microscopic invasion (CVC measurements).

Research Variable	Free CTC (Cell/mL)	*p*-Value	Cluster/mL	*p*-Value	Total CTC (Cell/mL)	*p*-Value
Vascular invasion	304.9 (151.6–656.4)	0.662	12.1 (3.1–29.5)	0.368	421.8 (162.7–697.9)	0.692
No vascular invasion	291.8 (116.1–480.9)		26.6 (4.3–41.0)		384.4 (123.3–522.5)	
Lymphatic invasion	304.9 (79.1–455.7)	0.504	31.1 (2.7–42.6)	0.565	421.8 (87.8–501.6)	0.629
No lymphatic invasion	291.8 (133.1–582.6)		13.3 (3.8–31.0)		384.4 (161.0–627.4)	
Neural invasion	350.5 (84.9–498.5)	0.728	13.4 (2.7–42.1)	0.667	443.1 (86.8–537.7)	0.728
No neural invasion	275.8 (164.5–625.3)		14.5 (10.5–32.8)		384.4 (198.2–668.0)	

**Table 7 cancers-13-06153-t007:** Correlation of CTC and CTC clusters with microscopic invasion (PV measurements).

Research Variable	Free CTC (Cell/mL)	*p*-Value	Cluster/mL	*p*-Value	Total CTC (Cell/mL)	*p*-Value
Vascular invasion	321.7 (217.4–451.3)	0.104	16.3 (5.5–41.5)	0.362	374.2 (271.0–508.7)	0.089
No vascular invasion	184.5 (97.2–322.3)		12.3 (3.2–29.9)		247.2 (125.4–401.0)	
Lymphatic invasion	204.7 (74.0–514.3)	0.489	23.8 (5.8–52-5)	0.397	350.5 (88.9–635.6)	0.939
No lymphatic invasion	248.2 (118.2–377.3)		12.3 (4.3–31.3)		305.5 (138.2–419.2)	
Neural invasion	247.3 (94.3–375.3)	0.905	13.8 (5.5–34.5)	0.528	327.3 (120.6–492.4)	0.798
No neural invasion	222.5 (113.0–370.8)		10.3 (3.1–32.2)		285.2 (141.1–404.7)	

**Table 8 cancers-13-06153-t008:** Median survival by groups (months).

Research Variable	Deaths N (%)	Median Survival	95% CI	Log-Rank-Value
<185 portal CTC	5 (38.5)	24.5	19.6–29.4	0.018
≥185 portal CTC	14 (63.6)	10.0	7.4–12.5	
<15 portal clusters	8 (42.1)	19.0	15.8–22.2	0.040
≥15 portal clusters	11 (68.8)	10.0	7.2–12.8	
No vascular invasion	9 (40.1)	22.5	17.7–27.2	0.005
Vascular invasion	10 (76.9)	10.0	5.3–14.7	
No lymphatic invasion	11 (42.3)	18.0	15.6–23.8	0.044
Lymphatic invasion	8 (88.8)	10.0	5.6–14.4	
No neural invasion	7 (41.2)	21.5	15.5–27.4	0.249
Neural invasion	12 (66.7)	13.0	4.9–21.0	
Degree of differentiation				0.003
G1–G2	15 (53.5)	19.0	4.6–23.1	
G3	2 (40.0)	16.0	7.7–24.7	
Global survival		18.0	12.5–23.5	

**Table 9 cancers-13-06153-t009:** Prognostic factors for long-term survival. Covariates were patient age, disease stage (I–II), portal CTC ≥ 185, portal clusters ≥ 15, microscopic invasion (vascular, lymphatic, neural), and tumor degree of differentiation (G1–3).

Research Variable		Univariate			Multiple	
	HR	95% CI	*p*-Value	HR	95% CI	*p*-Value
Sex (male)	2.050	0.789–5.325	0.140			
Age	1.000	0.949–1.054	0.988	-		
Stage I vs. II	1.566	0.555–4.415	0.396	-		
≥185 portal CTC	3.236	1.135–5.221	0.028	4.464	1.316–15.152	0.016
≥15 portal clusters	2.486	0.989–6.247	0.053	1.330	0.443–4.528	0.624
Vascular invasion	3.568	1.346–9.457	0.011	3.663	1.321–10.204	0.013
Lymphatic invasion	2.418	0.969–6.031	0.058	2.512	0.940–6.711	0.066
Neural invasion	1.698	0.666–4.328	0.268	-		
G1–2 vs. G3	0.841	0.196–3.771	0.841	-		

CI: confidence interval; HR: adjusted hazard ratio.

## Data Availability

The data presented in this study are available on request from the corresponding author.

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
