# Peer review of "Circulating Tumor Cells Enumeration from the Portal Vein for Risk Stratification in Early Pancreatic Cancer Patients"

_cancers, 2021, doi:10.3390/cancers13246153_

Round 1
Reviewer 1 Report
I appreciate the authors’ effort in addressing the issues and concerns raised by the reviewers. I still have several minor comments about the current version:
- Page 2, “the study included stage I and II of disease”. Please add numbers of stage I and stage II patients. Table 9, please indicate that it is stage I vs. II, or stage II vs. I.
- Why wasn’t gender included in Table 9?
- Please carefully read through the manuscript and correct all the typos and errors. For example, in page 13, the authors wrote “…in all patients en both samples. Differents combination…”.
Reviewer 2 Report
In their revised manuscript entitled „Circulating tumor cells enumeration from portal vein for risk stratification in early pancreatic cancer patients”, Padillo-Ruiz and colleagues are able to alleviate some of my previous concerns, mostly related to the information of the performed experiments. Some very important points, concerning adequate controls and sequencing results to undoubtedly demonstrate the cancerous origin of the detected cells are still lacking.
The authors still fail to discuss some very important and undoubtedly interesting results. Most importantly, why do they detect higher rates of CTC/ml in the CVC blood, than in the PV blood? As the authors point out, the PV is so much closer to the “seeding” tumor. Thus, the concentration of CTCs should be significantly higher there, as has been shown in the literature the authors themselves cite. If one considers the venous blood flow from the pancreas, the observation of increases CTC numbers in the CVC compared to the PV blood irrespective of detection method or target, as described herein, is hard to fathom and should be further elucidated.
The CVC is correctly placed within the superior caval vein before, not inside the right atrium, as the authors correctly point out by “hinted into the right atrium” (see, for example, PMID: 10992822). The superior caval vein collects all venous blood from the upper half of the body. The cells detected there will thus have left the tumor, travelled through the portal vein and the sinusoids of the liver (first capillary bed), then through the lower caval vein into the right heart; from there, they would pass through the lungs (second capillary bed) and into the left ventricle. Assuming the cells survived this long, they would pass in the arterial high blood pressure system and be dispersed throughout the whole body, some reaching the upper half, passing through a third capillary bed in various organs (i.e. the brain, muscles in the arms, etc.) to be collected in the superior caval vein and thus reaching the tip of the CVC. In light of this, the authors should definitely discuss why they find more “CTCs” and clusters in the CVC samples than in the PV samples, opposed to all other previously published results and current understanding, and what mechanism they suppose is the cause of this.
As specified in the previous review of the manuscript, blood samples (CVC and PV) from non-cancerous patients undergoing pancreatic surgery (i.e. for chronic pancreatitis or even pre-malignant lesions (IPMN)) are needed as negative controls to demonstrate that the detected cells are not purely surgery-associated. This important control is still missing in the revised manuscript. Regarding the “healthy controls” (n=3, which is also an insufficient amount), was the blood sample also collected from a CVC? Or was peripheral venous blood sampled by venipuncture?
Further, the use of a commercial kit (here, the “Isoflux™ Epithelial to Mesenchymal Transitions Circulating Tumor Cell Enrichment Kit”) does also not justify omitting the targets of the included antibodies, which the authors still do not list. According to this reviewer’s information, the kit utilizes both EpCAM- and EGFR-antibodies for the detection of epithelial cells, as well as N-Cadherin and Vimentin as mesenchymal markers. This information, if correct, should be included in the manuscript.
I agree with the authors that the clinical value of this work is the defined cut-off for a number of detected cells and the ability to predict the OS based on this. Yet scientifically, I still differ on the assumption that the majority of the cells detected as “CTC” and “clusters” here are indeed that. As mentioned previously, should obtaining blood samples from patients undergoing surgery for benign causes prove difficult, single-cell level molecular analysis (for example k-ras mutations, which are abundant in PDAC) of the “CTC” and “clusters” detected would also help to dispel doubts about the provenance of the detected cells. The resolution needs to be single cell level, because undoubtedly there will be some CTC within the hundreds of detected cells in this experimental setup and using, for instance, ddPCR to detect k-ras mutations in bulk samples would not shed any more light on the ratio of “true CTCs” vs. “false positives” in this manuscript.
Minor points:
Please carefully re-read the newer, yellow-marked paragraphs and potentially enlist the help of a native speaker, as some sentences are grammatically very challenging.
Please refrain from over-interpreting non-significant results as “trend” (p9) or “could be statistically substantial” (p14, 3rd paragraph) if the numbers were higher. This is not a scientific interpretation of the available data.
Please rephrase the last paragraph on p14, as it does not make any sense (“The high number…”).
Reviewer 3 Report
The authors have addressed my comments and subtantially improved the manuscript. It is much clearer now. Many thanks
Round 2
Reviewer 2 Report
First off, I would like to thank the authors for an interesting and novel manuscript and the challenging work they have done. They were able to address some of my concerns and have further improved their manuscript in the process. Never the less, the authors have still not provided the answers I would like before recommending their work for publication.
As stated twice before, the authors use a so far unproven mix of antibodies that detect both mesenchymal and epithelial markers in central venous and portal vein blood of pancreatic cancer patients. As negative control, they use the blood of three healthy, non-operated volunteers. (The authors also did not answer the question whether the blood was from the CVC or the PV of healthy volunteers.) Again, three non-operated volunteers is not an adequate control in quality or quantity. The patients in this study are undergoing major surgery, with significant tissue damage and thus myriad possibilities of both epithelial and mesenchymal cells to enter the bloodstream. In light of the results diverging from the literature regarding the huge number of detected “CTC” in the PV and CVC blood, intraoperative blood samples from a pool of non-cancerous patients are needed as negative controls to demonstrate the specificity of the method. If the authors do not detect “CTC” and “clusters” in these samples, that would demonstrate that the cells detected in PDAC patients are indeed CTC.
I still feel that the mentioned thorough controls are warranted before publishing this manuscript.
Author Response
Please see the attachment

This manuscript is a resubmission of an earlier submission. The following is a list of the peer review reports and author responses from that submission.
Round 1
Reviewer 1 Report
Padillo-Ruiz et al. determined CTCs and clusters in portal blood (PV) and central venous catheter (CVC), and explored their prognostic value in early pancreatic cancer. They found that the determination of CTCs and clusters in PV was better than in CVC as a prognostic marker in patients with early-stage adenocarcinoma in the head of the pancreas. Currently, only a few studies evaluated CTCs (and clusters) in portal and peripheral venous blood samples collected simultaneously during pancreaticoduodenectomy. The present study focused on early (stage I/II) pancreatic cancer sited in the head (N=35), and again demonstrated the prognostic value of CTCs in portal vein from PDAC patients. They also observed significantly shortened OS in patients with high portal clusters, although the association was not independent of clinical confounders. The following issues need to be addressed:
- Figure1: image of CTC in CVC shows positive CD45. How to explain?
- How about the recovery rate of CTC detection?
- How many patients without CTC or cluster in PV and CVC?
- Tables 1, 2, 3 6, 7: please use median with range to take the place of mean±SD for CTC enumeration results and test statistical difference between medians.
- Page 5, how did the authors establish the cutoffs of cluster number (20 clusters/ml) and CTC inside a cluster (64 CTC/cluster) for predicting a G3 PDAC? Page 6, why set up the cutoffs according to the 33rd percentiles? Were the cutoffs of 185 CTCs and 15 clusters applied in both PV and CVC samples?
- Table 8: please add the numbers of total patients and numbers of patients who died for each subgroups.
- Table 9: please indicate the reference group when stage was analyzed as a prognostic factor?
- Figures 2 and 3 can be combined into one figure with two panels.
- Please add a figure to show the Kaplan-Meier curves for disease-free survival analysis.
- Inconsistent description: ten ml of PV and CVC blood in the Abstract, but seven milliliters of whole blood samples in page 2. Page 3, the statement “while the cluster number was higher in CVC than PV” was different from the data in Table 1.
Reviewer 2 Report
In their manuscript entitled „Circulating tumor cells enumeration from portal vein for risk stratification in early pancreatic cancer patients”, Padillo-Ruiz and colleagues report on the enumeration of circulating cytokeratin positive cells in 35 surgical patients suffering from pancreatic cancer. Using the Isoflux system and a commercial antibody kit (“EMT CTC kit Izasa”) to coat unspecified beads, following a density-centrifugation step, the authors describe successful detection of mean >300 cells in the intraoperatively sampled central venous (CVC) and portal vein (PV) blood. The cells were further stained for pan-CK, CD45 (both AB not specified) and a nuclear dye (see below). The authors define these CK+/CD45- nucleated cells as circulating tumor cells and further report on the detection of cell clusters. Using an automated image counter based on the detection of circular form (as published by the authors previously), the authors correlate the number of cells and clusters with the sampling site (central venous vs portal blood) and clinical characteristics.
The authors describe a significant increase in “free CTC”, “cluster” and “total CTC” number in the central venous blood, compared to the intraoperatively sampled portal vein blood. The detection of “clusters” and “CTC inside a cluster” further correlated with lower differentiation in the PV blood, but not in the CVC blood. There were no further correlations between clinicopthological characteristics and the detected CK-pos cells. The OS, but not the DFS, was significantly longer in patients with less than 185 cells, or less than 15 clusters, in the PV sample. The same was not found for CVC blood samples. In the multivariate analysis of PV blood samples in stage I&II vs stage III patients, only less than 185 cells and vascular invasion remained significant prognostic markers.
While the authors present a well-written manuscript with sound statistics, there are many major concerns with this work:
First, the manuscript lacks important information regarding the methodology. The authors give no information on the antibodies used, the experimental settings (i.e. density centrifugation, Isoflox settings, precise antibody-mix, magnetic beads, etc.) or any concentrations. These are fundamental information for any research paper that simply cannot be omitted.
Second, the authors do not include any controls or references for the applied system. While the magnetic beads based Isoflux system has previously been used to isolate CTC, the unspecified antibodies conjugated to the also not specified beads (Dynabeads?) used here are unproven. The authors should provide controls in the form of blood samples from healthy donors and patients undergoing pancreatic surgery for non-malignant pancreatic disease to prove the specificity of their system. Are the “CTC” detectable in equally high frequencies and numbers in these control patients? Additionally, titration experiments with spiked pancreatic cancer cells in-vitro are needed to assess the sensitivity of their assay.
Third, the use of a nondisclosed “cocktail of antibodies” to define circulating cells from patients undergoing a major surgical intervention as “circulating tumor cells” is insufficient. It has previously been demonstrated that during surgery, more cells may be dislodged and thus be detectable in PV blood (33415562). Additionally, endothelial cells may also have been detected by the unspecified “cocktail of antibodies including epithelial and mesenchymal markers” used in the current study (34455700). The authors need to provide sequencing results demonstrating tumor-specific mutations in the detected cells, preferably on a single cell level.
Fourth, the number of detected “CTC” reported here are extremely high (mean of >300/ml!). Previously, using the FDA-approved and thoroughly tested CellSearch System (using EpCAM and CK), the number of CTCs in 7.5ml of PDAC patient blood have consistently been in the low single digits in the peripheral, as well as the portal vein blood [just a few recent Refs: 33513877, 33415562, 31717747). In addition, in the current study, CTC were detected in all patients (100%) in both PV and CVC blood. This is in stark contrast to the usual reported rates of ~10% for peripheral blood and below 25% in PV-blood of resectable PDAC patients (33513877, 33415562, 31717747). Also, the fact that the current study found more CTC in the central venous blood than in the portal vein blood is contradictory to previous reports, which consistently demonstrated a higher concentration in portal vein blood (33415562, 31717747). This higher concentration is, as the authors’ themselves point out, easy to explain if one considers the venous drainage of pancreatic tumor through the portal vein. All this indicates, to this reviewer, a highly unspecific detection of CK-positive cells in the bloodstream of patients undergoing major surgery, which may originate from the substantial tissue damage resulting from the operation. The argument that the smaller cell-clusters detected in the CVC blood may be due to the clusters breaking up by passing through the liver is flawed, as there are also less “total CTC” in the PV blood. As all CTC originating from the PDAC will need to pass through the PV and the liver to reach the CVC, one would expect a higher “total CTC” count in this compartment (PV) and maybe a shift in the clustered-to-free ratio in the CVC blood. The observed higher absolute number of CTC in the CVC blood is simply not explained by the break-up of clusters.
Fifth, the timing of the blood draw in relation to the surgery needs to be specified and should be consistent throughout the whole study. Was the blood collected during surgery? Before manipulation of the tumor? Following dissection of the hepatoduodenal ligament? Was the portal vein dissected or was it punctured trans-hepatically? Was the first blood-draw discarded to exclude epithelial cells dislodged by the venipuncture? All these factors play a role for the frequency of epithelial cells in the PV- as well as in the CVC-blood, as these steps may lead to unspecific shedding of epithelial cells.
Minor points
Was Hoechst used to counterstain, as described in the text (line 03), or was DAPI used, as indicated in Figure 1?
Please correct “Citokeratin” in the heading of Figure 1.
Please define “early” pancreatic cancer.
Please specify the proportion of R0, R1 and R2 resected patients and include only curatively resected (R0/1) patients in the survival analysis.
Please elaborate on the new “Panic-1” cell line used in-vivo (line 88).
Please provide the total number of included patients (“n=XX”) in the survival analyses (Fig 2,3, Cox-reg.)
Was the tumor differentiation defined in the “biopsy” (line 226) or in the resected specimen?
Reviewer 3 Report
Circulating tumor cells enumeration from portal vein for risk stratification in early pancreatic cancer patient
1- Please define G1-G3 mentioned in the abstract, undefined abbreviations in the abstract are not advised.
2- In PV, the CTC number per cluster was higher in patients with G3 than in patients with G1-G2 (52.9±55.2 vs. 128.4±163.8; p=0.05). What is the reason for this?
3- p in p-value must be italicised throughout.
3- The number 185 as a cut-off is very specific, is there a STDEV?
4- The number of CTC and clusters correlates with the differentiation degree of pancreatic adenocarcinoma. Please indicate if this correlation is positive or negative?
5- Good point about the PV circulation from the GI tract to the liver, although potentially a PV blood sample may be clinically invasive? Also, the study does find that the PV CTCs are correlated with prognosis and survival and not the CVC. It would be interesting to know what the results would be for referral blood (e.g., referral to the literature).
6- It is custom to highlight the most important aspects of the findings of the paper in the last line of the abstract, but it is up to you.
7- “pancreatic cancer sited in the head”, please consider rewording this since it is vague.
8- Please spell all acronyms when they first appear in the text including FDG-PET-CT.
9- So were patients with overt metastasis not included in the study? Only the patients with no CT metastasis or those with suspicious lesions? I was wondering what the logic behind this exclusion is? Wouldn’t it be better to include all this range, to determine how the number of CTCs differ in this spectrum even though you are assessing early disease/ MRD/ no metastasis since this might provide more meaningful and significant CTC/ cluster cut-offs? Given the inclusion and exclusion criteria were all the original patients mentioned included (or are the 35 patients post-selection?)
10- Was there a correlation between the location of invasion and degree of differentiation?
11- Did all patients receive gemcitabine or only a fraction?
12- Please define CK again in section 2.2, also PMBC.
13- Some English editing is required
14- Cocktail of antibodies include epithelial and mesenchymal markers”. Please provide specific detail about what these markers were. Do you mean CD45 for stromal mesenchymal and CDK for epithelial?
15- Was the Isoflux designed to positively or negatively select? Similar to MACS, is it selecting positive markers and enriching them? Specifically, what were the output cells? The exact input, isolation logic, and output need more explanation.
16- “CTC are enumerated as morphologically intact CK+/CD45-/nucleated cells”. Again, this needs to be clarified in light of the isolation logic mentioned in the comment above. Also, EMT is involved in metastasis, so it would be interesting to understand the nature of these CTCs and why they have downregulated stromal markers and upregulated epithelial markers. Please provide support from the literature about markers of pancreatic CTCs.
17- “Although we observed that the overall CTC detection was higher in CVC than in PV, there were no statistically significant differences”. That is an interesting result since anatomically speaking one would expect the PV to have higher CTC and cluster counts. Also even though you have counted fewer CTCs in the PV, they correlate with OS and prognosis.
18- Also please add some detail about the characteristics of CTCs and clusters in pancreatic cancer.
19- “While the cluster number was higher in CVC than PV, the size showed an inverse correlation. The cluster size was more increased in PV than in CVC”. This is probably due to mobility proportional to size.
20- Figure 1, were any attempts made to quantify the positive cells in the microscopy? If so, I assume the difference was not statistically significant akin to previous results?
21- In Table 1 please annotate the 2 p-values that appear.
22- What is the significance of a ratio of Free CTC to inside a cluster that is significantly different between PV and CVC?
23- Table 2, no correlation between differentiation degree and CTC/ clusters. What does this mean?
24- I think the authors have forgotten to mention (or refer to) the data in Tables 2 and 3 in the main text. At any rate, there is no correlation with the degree of differentiation (apart from one that is borderline; 0.05).
25- “On the one hand, to predict a G3 pancreatic adenocarcinoma presence, according to these results, we established a cut-off for cluster number of 20 clusters/ml and CTC inside a cluster of 64 CTC/cluster.” It is not clear how this cut-off has been established, is this based on tables 2 and 3 or other data, please explain.
26- In tables 4 and 5, please explain, what a tumour size of -0.1 means. There will be many readers of this paper, who will not be familiar with a lot of these terms and concepts, so kindly explain. The same applies to Rho is Ca19.9, the metrics for this marker require explanation.
27- I would advise against used contractions “didn’t”.
28- “Similarly, there was no significant correlation between CTC measurements with vascular, neural, or lymphatic invasion”. What is the justification for this? In general, this manuscript does not sufficiently interpret and explain results.
29- In Table 8, please replace median with “median survival”.
30- How was the 33rd percentile established resulting in a cut-off point of 185 for CTC and 15 for clusters?
31- “Regarding PV measurements, the median patient’s sur- survival was significantly longer in patients with less than 185 CTC (24.5 months [IC 19.6- 29.4] vs. 10 months [CI 95% 7.4-12.5]; p=0.018) and less than 15 clusters (19 months [CI 95% 15.8-22.2] vs. 10 months [CI 95% 7.2-12.8]; p=0.004). Patients with vascular (p=0.005) or lymphatic invasion (p=0.044) were associated with less survival. ”I think this is the highlight of your study. Do the authors have any justification for why the neural invasion was not significant? What is the difference from a physiological viewpoint?
32- Figures 2 and 3 require much more detailed figure legends. Kindly explain what the red and blue lines represent (i.e., high/ low expressing groups). If this is a Kaplan Meier curve then kindly mention the statistics, the HR values, and any relevant values for these figures in the legends.
33- [CI 95% 7.2-12.8]; p=0.004). (Table 8 and Figures 2-3). There is a full stop between the brackets that do not belong there. Also, errors in (HR=4,464; p=0,016) and no vascular invasion (HR: 3,663; p=0,013), comma use is incorrect.
34- Table 1 legend says (I and II versus III), while the actual table says I and II. Also please use proper annotations for p-values. What do the asterisks in the table refer to?
35- “The multivariable analysis showed that both the number of CTC<185 (HR=4,464; p=0,016) and no vascular invasion (HR: 3,663; p=0,013) were independent predictors of better long-term survival (Table 9)”. What is the justification for clusters not being independent predictors while CTCs were?
36- The data reported in 3.3. are not shown in a table. Lymphatic invasion appears to be significant in multiple survival and prognostic tests. Can the authors explain this in the text?
37- “As in other studies [7,22], the disease-free time in metastases appearance and local recurrences was lower in patients with values above the cut-off in CTC and clusters, although the differences were not statistically significant. Thus, we could not conclude that in our study CTC and cluster levels are predictors of metastasis or local recurrence in the follow-up of patients”. This may be different if advanced stage and highly metastasised disease CTC and clusters were compared with no metastasis, but your aim was testing no metastasis samples.
38- “However, related to OS, patients with more than 185 CTC and 15 clusters in PV had significantly lower survival. In the multi-variate study, CTC>=185 in portal blood (HR 4.4) and vascular invasion (HR 3.6) were independent predictors for survival. These findings were not observed in CVC determinations.” It would be interesting to analyse the physiological difference between PV and CVC, which influences significant association with survival, and multivariate studies for predictors of survival. As the authors suggest, the liver first pass may eliminate cells/ cell clusters by macrophages or any activated T cells.
